# Association between serum mineral concentrations and gastrointestinal parasite burden in zebu cattle accessing '*hora*' mineral water in southwestern Ethiopia

**Ashenafi Miresa**[1]*, **Taye Tolemariam**[1], **Belay Duguma**[1], **Ellen S. Dierenfeld**[2], **Abebe Nigussie**[3], **Feyissa Begna**[4]

**1** Department of Animal Science, College of Agriculture and Veterinary Medicine, Jimma University, Jimma, Oromia, Ethiopia, **2** Department of Veterinary and Biosciences, Faculty of Veterinary Medicine, Ghent University, Ghent, Belgium, **3** Department of Natural Resource Management, College of Agriculture and Veterinary Medicine, Jimma University, Jimma, Oromia, Ethiopia, **4** School of Veterinary Medicine, College of Agriculture and Veterinary Medicine, Jimma University, Jimma, Oromia, Ethiopia

* ashenafi.miresa@yahoo.com

## Abstract

Gastrointestinal parasites (GIP) and mineral deficiencies are significant factors affecting health and productivity of free-ranging cattle. Adequate mineral intake, particularly from natural mineral water sources (hora), is vital for immune function, gastrointestinal health, and nutrient absorption. This study aimed to explore the association between GIP burden and serum mineral concentrations in zebu cattle (Bos indicus) routinely accessing hora mineral water in southwestern Ethiopia. A total of 180 fecal samples were collected from cattle across four districts (Bedele, Dabo, Gechi, and Borecha) and analyzed qualitatively and quantitatively to determine parasite presence and fecal egg count. Concurrently, blood samples were collected to evaluate serum mineral concentrations. The overall GIP prevalence was 55.6%, with Strongyle-type nematodes being the dominant GIP group. Gechi district showed the highest prevalence (64.4%) and mean egg per gram (EPG) of 212.8 ($p < 0.05$). Although, serum mineral concentrations were generally adequate, significant variations were observed across districts. Strong negative correlations ($p < 0.05$) were observed between EPG and serum concentrations of zinc (Zn), manganese (Mn), iron (Fe), and copper (Cu), indicating that adequate intake of these minerals, potentially sourced from the hora mineral water, may be associated with improved resistance to parasitic infections. These negative correlations were supported by negative binomial regression analysis which identified Zn as the strongest predictor of EPG. Overall, the findings highlight the importance of hora as a natural mineral supplement in its association with lower GIP burden in free-ranging zebu cattle. While this study indicates a correlation between serum mineral concentrations and GIP burden in

**Data availability statement:** "All relevant data are within the paper and its Supporting information files."

**Funding:** This research was supported by Jimma University College of Agriculture and Veterinary Medicine (grant number AgVmVm/M6/23) which provided funding for data collection and laboratory analysis. However, the funders had no role in study design, data collection and analysis, decision to publish, or preparation of the manuscript.

**Competing interests:** The authors have declared that no competing interests exist.

grazing cattle, controlled experiments are essential to determine the specific effects of individual minerals found in hora on parasite resistance and establish causality.

## Introduction

Cattle play a significant role in global agriculture, providing essential resources such as food, draft power, and income for millions of rural households, especially in developing countries [1–3]. Zebu cattle (*Bos indicus*), particularly the *Horro* breed, are central to the livelihoods of rural communities in southwestern Ethiopia, providing vital resources such as milk, meat, manure, and draft power [4,5]. The *Horro* breed is well known for its adaptability to local conditions making it prevalent and important breed in this region [6]. However, their health and productivity are often compromised by various factors, including GIP infections and mineral deficiencies [7,8]. GIPs pose a significant threat to cattle health, impairing immune function, hindering nutrient absorption, and ultimately reducing weight gain and productivity [9]. These parasites can also exacerbate mineral malabsorption, creating a negative feedback loop that further weakens the animals [10,11]. GIPs such as nematodes, trematodes, and cestodes are a major global concern in cattle production and cause substantial economic losses through diminished livestock productivity and compromised animal health [8,12,13]. Essential minerals such as calcium (Ca), phosphorus (P), magnesium (Mg), iron (Fe), manganese (Mn), copper (Cu), and zinc (Zn) are essential for maintaining cattle health and enhancing immunity [14,15]. Deficiencies in these minerals can compromise immune responses, making animals more susceptible to infections and diseases [16,17]. Specifically, trace minerals such as Zn, Cu, Se, and Fe have been shown to have direct anthelmintic effects by enhancing immune function and reducing oxidative stress [18,19]. Moreover, research indicates that zinc and copper play significant roles in lowering gastrointestinal parasite burdens in naturally infected sheep [20].

Natural mineral sources, including mineral-rich soils and mineral water sources locally known as "*hora*", are traditionally used as mineral supplements for livestock in Ethiopia [21–23]. In southwestern Ethiopia, *hora* is a valuable sources of mineral supplementation for grazing cattle, potentially providing them with adequate mineral intake. It is believed to provide essential minerals to grazing livestock [24,25], including (but not limited to) Na, K, Ca, Mn and Zn [22]. It can be a critical resource for cattle in this region, especially given the potential for mineral-deficient soils and pastures. While previous studies have highlighted the role of minerals like Zn, Cu, and Se in enhancing immune function and potentially reducing parasite burdens [25,26], there is limited understanding of the direct relationship between serum mineral concentrations and GIP burden in zebu cattle that access *hora* mineral water. This suggests a critical knowledge gap, as understanding the relationship between serum mineral concentrations and GIP burden can inform the development of targeted mineral supplementation strategies and optimization of *hora* use for improved cattle health and productivity.

Therefore, this study aims to explore the association between GIP burden and serum mineral concentrations in zebu cattle routinely accessing *hora* mineral water in southwestern Ethiopia. We hypothesize that among cattle accessing *hora* mineral water, higher serum concentrations of essential minerals will be negatively correlated with GIP burden. Serum mineral concentrations will reflect the animal's mineral status [27,28], while fecal samples will allow for the quantification of parasite burden [29,30]. By analyzing both blood serum and fecal samples, this research contributes to a better understanding of the nutritional status of zebu cattle in this region and the potential role of *hora* as a natural mineral supplement in improving cattle health and productivity.

## Materials and methods

### Study sites

The study was conducted from October to December 2023 in Buno Bedele zone, situated in the southwestern Ethiopia (Fig 1). Among the nine districts of Buno Bedele Zone, four districts—Bedele, Dabo, Gechi, and Borecha—were purposively selected based on the availability of *hora* mineral water. These districts are predominantly located in mid- to lowland agroecological zones and share similar weather conditions. Across these selected districts, smallholder farmers utilize 'hora' as a mineral supplement for free-ranging cattle.

### Sample collection

The current study used indigenous *Horro* breed of zebu cattle (*B. indicus*) which are prevalent in the study area and regularly access *hora* mineral water as a mineral supplement. In each district, 45 sexually matured male and mature dry female (in the age range of 5–7 years) cattle were selected from 9 herds, totaling 180 animals (n = 180). To ensure a representative sample and minimize bias, a systematic random sampling approach was employed at the animal level within selected herds. This involved, selecting animals by pointing randomly to individuals within the grazing herd from a fixed vantage point, ensuring avoidance of conscious selection bias. Sample size was determined based on the recommendations for both mineral status assessment (15–20 animals; Ensley [28]) and endoparasite monitoring (7–20 animals; Maurizio et al. [31]), ensuring adequate representation for both analyses. Prior to sample collection, the willingness and verbal consent from the cattle owners was obtained. Throughout all sample collection procedures, the welfare of the animals was prioritized. All handling was performed gently by experienced personnel to minimize stress and discomfort.

### Fecal sample collection

Approximately 10g of fecal samples were collected per rectum from each selected animal using sterile gloves and aseptic techniques. Samples were placed in sterile, screw-capped plastic bottles containing 10% formalin as a preservative. Following collection, samples were stored at 4°C until laboratory analysis, adhering to the procedure described by Smith et al. [32].

### Blood sample collection

Blood samples were drawn from the jugular vein using 23 G needles (MN-2038M) with a non-additive vacutainer (10 ml, plain tube) and labeled with relevant details. To minimize hemolysis, samples were handled gently to avoid mechanical stress, and tubes were placed at a 4°C incline for 24 hours to slow the clotting process. Samples were visually inspected for hemolysis and any samples showing hemolysis were discarded. After complete coagulation, serum was carefully separated and transferred into 5mL cryovial tubes. All serum samples were then stored at −20°C until laboratory analysis for mineral profile determination.

### *Hora* mineral water collection

Concurrently with fecal and blood sample collection, *hora* mineral water samples were collected from the sources used by cattle in each district. Samples were collected in sterile bottles and stored at 4°C until mineral analysis. The

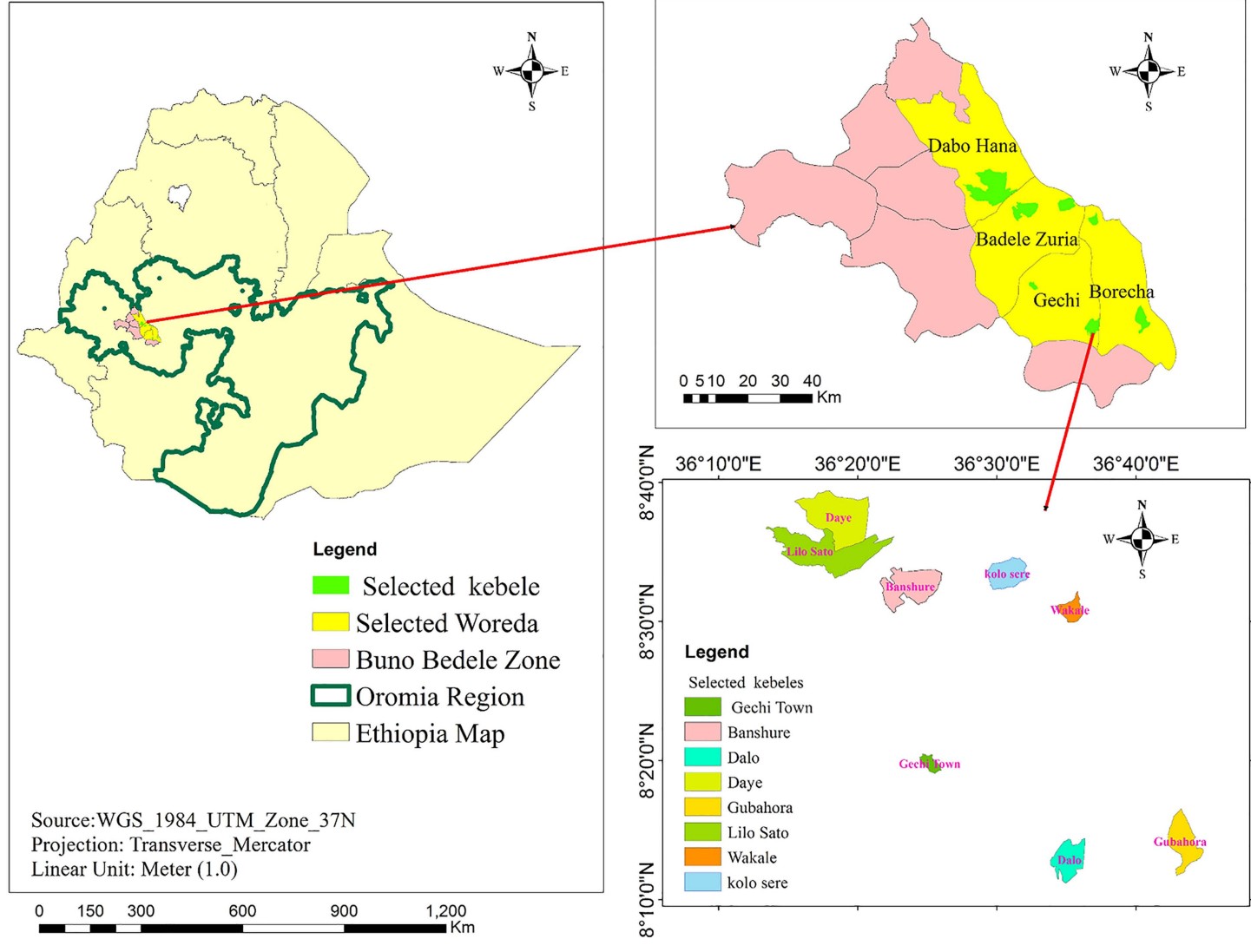

**Fig 1. Map of the study area.**

physicochemical properties and mineral concentrations of the *hora* water were analyzed and the raw data for this analysis is provided in S3 File, whereas the statistical output and results are provided in the Supporting file (Appendix 3 in S1 File).

### Coprological examinations

**Qualitative analysis.** Fecal samples were subjected to both floatation and sedimentation techniques. The simple floatation technique, using a saturated sodium chloride solution (400 g NaCl/1000 mL of distilled water, specific gravity 1200), was employed for the detection and initial morphological classification of nematode and cestode eggs, following established diagnostic guidelines [33,34]. Briefly, 5g of fecal sample was crushed, suspended in the solution, sieved, and transferred to 10 mL floatation bottles. Coverslips were applied to the surface of the solution and allowed to stand for 10 minutes. The coverslips were carefully picked and subsequently mounted on glass slides and examined under a microscope at 10x magnification to locate potential parasite

and switched to 40x magnifications for detailed observation. The sedimentation technique was used to identify trematode eggs, which are denser and heavier than other parasite eggs and performed according to FAO [35]. A 3g fecal sample was mixed with 30 mL water in a beaker, then filtered through a tea strainer or double-layer of cheesecloth into a test tube. The filtered material was then allowed to sediment for 5 minutes. After this first sedimentation, the supernatant was carefully removed. The sediment was then resuspended in 5 mL of water and allowed to sediment for a second period of 5 minutes. Following this, the supernatant was again carefully discarded. The final sediment was stained by adding one drop of methylene blue, transferred to a microslide, and examined under a dissecting microscope.

**Quantitative analysis.** For nematode and cestode eggs, positive samples were subjected to quantitative analysis using a modified McMaster technique as described by Cork and Halliwell [36]. For trematode eggs, a quantitative analysis was performed using a Modified Stoll's dilution technique. Nematode egg with similar morphology were classified as 'Strongyle-type'. On the other hand, eggs of *Trichuris spp.*, *Ascaris spp.*, *Strongyloides spp.*, trematodes, and cestodes were identified to the genus level based on their unique morphological characteristics.

## Serum minerals analysis

Serum samples were prepared for mineral concentration analysis by microwave digestion using the method described by Toniolo et al. [37]. Serum samples were thoroughly vortexed to ensure a homogenous matrix prior to digestion [38]. To prevent settling, samples were immediately pipetted after vortexing. Briefly, 250 µL serum aliquots were digested with 300 µL $HNO_3$, 200 µL HCl, and 100 µL $H_2O_2$. After digestion was completed (15 minutes), the samples were transferred to 15 mL Falcon tubes, and vessels rinsed with distilled water. Serum concentrations of Ca, Mg, K, Na, P, S, Fe, Zn, Mn, Cu, and Mo were analyzed by inductively coupled plasma-optical emission spectrometry (ICP-OES, Spectro Arcos, model FSH-12–2010; Spectro Analytical Instruments GmbH, Kleve, Germany) at Horticoop Ethiopia PLC. Analytical accuracy and precision were rigorously maintained throughout the study. Standard solutions were analyzed concurrently with samples to ensure calibration integrity. Reproducibility was assessed by calculating relative standard deviations (RSDs) and employing internal standards. Quality control measures included the regular analysis of control solutions and blanks to monitor for instrumental drift and contamination.

## Data analysis

All statistical analyses were performed using R software version 4.4.1 [39]. Non-detectable serum mineral concentrations were assigned a value of half the detection limit to allow for statistical analysis, following the approach recommended by Dermauw et al. [40]. Pearson Chi-squared test was used to assess differences in GIP prevalence (proportion of positive animals) among districts. Given that the distributions of serum mineral concentrations and fecal egg counts did not meet the assumptions of normality required for parametric tests, non-parametric statistical methods were employed. Spearman's rank correlation was used to assess the relationship between serum mineral concentrations and fecal egg counts (EPG). Following the identification of significant correlations, negative binomial regression model was used to further investigate the relationship between these variables. Negative binomial regression was chosen after an initial Poisson regression model showed evidence of overdispersion (i.e., variance greater than mean and theta greater than 1), which is common in count data (EPG). First, univariate negative binomial regression models were fitted for each serum mineral. Subsequently, a full multivariable model was constructed including all minerals as predictors, and stepwise regression using the Akaike Information Criterion (AIC) was then performed to select the most parsimonious model. Kruskal-Wallis tests with pairwise Wilcoxon rank-sum tests (Benjamini-Hochberg correction) were used to analyze differences in mean serum mineral concentration and EPG counts across districts. Statistical significance was declared at a probability level of $p < 0.05$ for all tests.

## Ethical approval

The experiment was reviewed for ethical clearance and approved (RGS/588/2023) by Jimma University, College of Agriculture and Veterinary Medicine, following the guidelines of the European Union directive number 2010/63/EU (2010) regarding the care and use of animals for experimental and scientific purposes.

## Results

### The overall prevalence of GIP in zebu cattle

GIP eggs were detected in 99 out of the 180 fecal samples collected, representing an overall prevalence of 55.0% (95% CI: 47.7% − 62.1%) in zebu cattle grazing around and routinely accessing *hora* mineral water (Table 1). The prevalence of GIP did not differ significantly among the study districts ($\chi^2 = 2.94$, $p = 0.40$). The raw data for gastrointestinal parasite prevalence are available in S2 File.

### Prevalence of specific parasite genera in positive samples

The prevalence of specific parasite genera and group was determined for the positive fecal samples for GIP and illustrated in Fig 2. *Strongyle-type* nematodes were the most prevalent group, accounting for 65.7% of the positive samples. Trematodes were less frequent, representing 14.2% of positive samples, with *Paramphistomum* spp. (6.1% of positive samples) and *Fasciola* spp. (8.1% of positive samples) being the predominant genera. *Moniezia spp.* was the only cestode identified, with a prevalence of 3.0% of the positive samples. *Trichuris spp., Strongyloides spp.*, and *Ascaris spp.* were also identified, as displayed in Fig 2.

### Egg per gram of GIP in zebu cattle

The mean fecal egg counts (EPG) of nematodes, trematodes, cestodes, and mixed parasite infections in zebu cattle across the study districts are presented in Table 2. The raw data for EPG of GIP in zebu cattle are available in S2 File. While no significant differences were observed in the mean EPG of each parasite type, the Gechi district consistently showed numerically higher EPG values. As a result, the overall mean EPG in the Gechi district was significantly higher ($\chi^2 = 11.92$, p = 0.01) compared to the Dabo and Borecha districts.

### Serum mineral concentrations of zebu cattle

The mean serum mineral concentrations of zebu cattle grazing around and accessing *hora* mineral water in southwestern Ethiopia are illustrated in Figs 3 and 4, and Supporting file (Appendix 1 in S1 File). The raw data for serum mineral concentrations of zebu cattle are available in S4 File. Serum Ca, Mg, and K concentrations were significantly higher in cattle from Borecha compared to Bedele, Dabo, and Gechi districts (p < 0.01). Similarly, Fe, Cu, and Zn concentrations were significantly higher in cattle from Borecha district (p < 0.01). In contrast, serum K concentrations were significantly lower

**Table 1. Prevalence (%) of gastrointestinal parasites in zebu cattle (*B. indicus*) accessing *hora* mineral water in southwestern Ethiopia.**

| Districts | Sample size | No of Positive | Prevalence (%) | 95% CI | χ2 | p-value |
|-----------|-------------|----------------|----------------|--------|-----|---------|
| **Bedele** | 45 | 25 | 55.5 | 41.2-69.1 | 2.94 | 0.40 |
| **Dabo** | 45 | 21 | 46.7 | 32.9-60.9 | | |
| **Gechi** | 45 | 29 | 64.4 | 49.8-76.8 | | |
| **Borecha** | 45 | 24 | 53.3 | 39.1-67.1 | | |
| **Overall** | 180 | 99 | 55.0 | 47.7-62.1 | | |

*χ2 = Pearson chi-squared; CI = Confidence interval.*

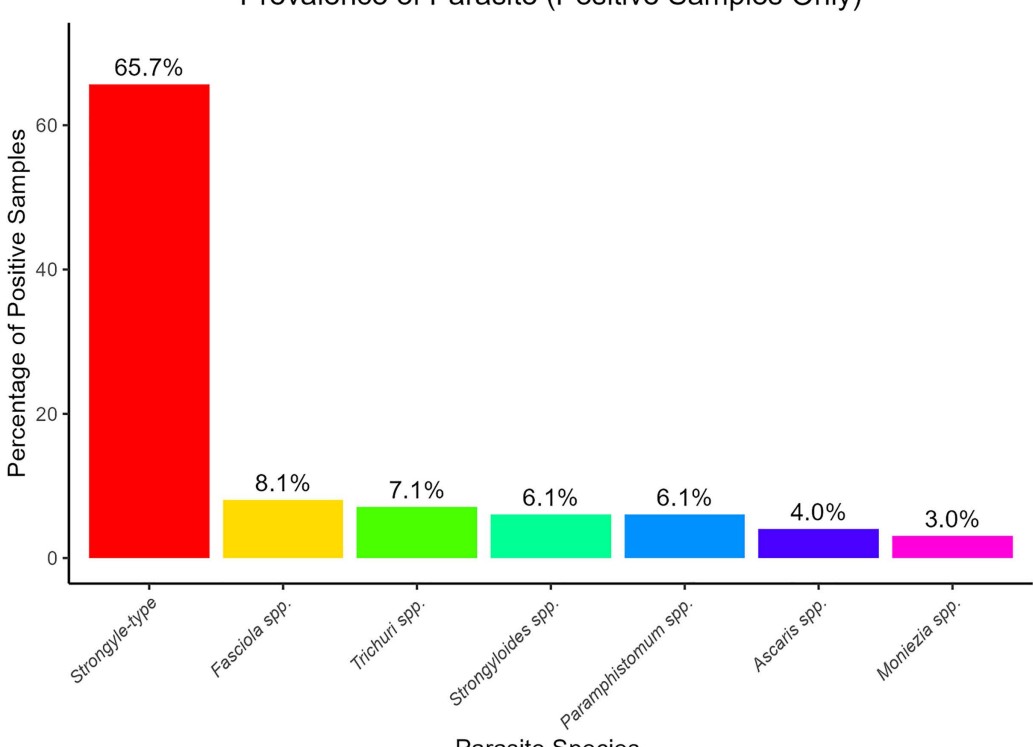

**Fig 2. Prevalence (%) of specific gastrointestinal parasites in positive fecal samples of zebu cattle (*B. indicus*) accessing *hora* mineral water in southwestern Ethiopia.**

**Table 2. Mean egg per gram (EPG) of gastrointestinal parasites in zebu cattle (*B. indicus*) in different districts of southwestern Ethiopia.**

| Parasite groups | Districts | | | | Overall | SE | $\chi^2$ | p-value |
|---|---|---|---|---|---|---|---|---|
| | Bedele | Dabo | Gechi | Borecha | | | | |
| Nematode | 103.20 | 84.02 | 125.56 | 93.60 | 101.59 | 11.06 | 1.50 | 0.68 |
| Trematode | 48.89 | 25.56 | 54.22 | 9.60 | 34.56 | 7.37 | 4.12 | 0.25 |
| Cestode | 21.29 | 4.22 | 7.40 | 10.38 | 10.82 | 4.13 | 1.71 | 0.63 |
| Mixed | 9.60 | 15.89 | 25.56 | 8.49 | 14.88 | 4.50 | 1.20 | 0.75 |
| Overall* | 182.97[ab] | 129.69[a] | 212.73[b] | 122.07[a] | 161.87 | 11.91 | 11.92 | 0.01 |

*Means in a row with similar superscripts are statistically similar; SE = Standard error; $\chi^2$ = Kruskal-Wallis chi-squared*

in cattle from Bedele district compared to Gechi and Borecha districts (p = 0.01). Furthermore, Mn concentrations were significantly higher in cattle from Dabo district compared to the Bedele, Gechi, and Borecha districts (p < 0.01).

### Correlation of serum mineral concentration with EPG of GIP in zebu cattle

Spearman correlation coefficients between serum mineral concentrations and mean fecal egg counts (EPG) in zebu cattle across the study districts are presented in Table 3. In Bedele district, negative correlations were observed between EPG and serum Fe (r = −0.76, p < 0.05), Cu (r = −0.92, p < 0.001), and Zn (r = −0.65, p < 0.05). Similarly, in Dabo district, negative correlations were observed between EPG and serum Fe (r = −0.98, p < 0.001), Cu (r = −0.89, p < 0.001), Zn (r = −0.94,

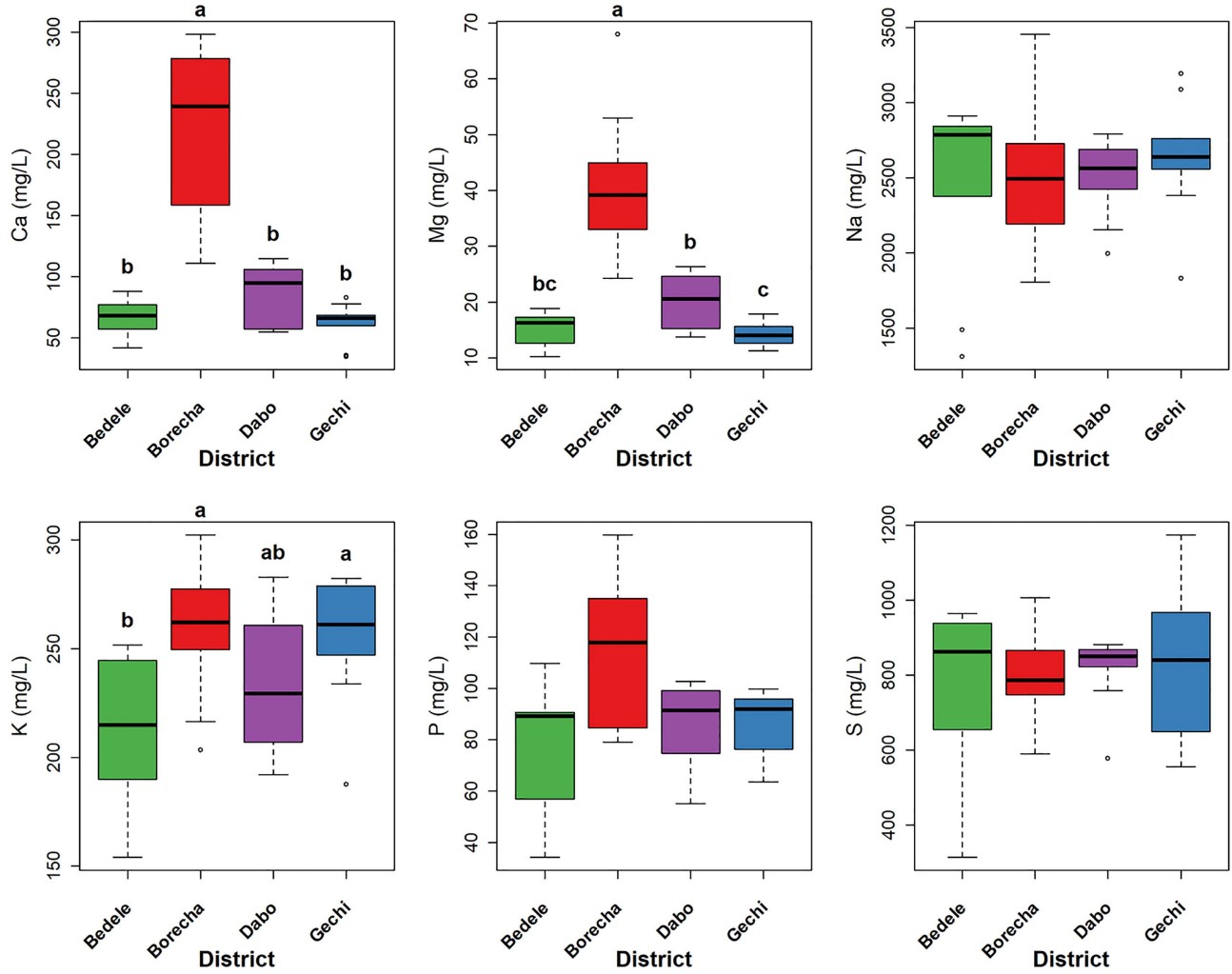

**Fig 3. Box plots of serum macro-mineral concentrations in zebu cattle (*B. indicus*) grazing around and accessing *hora* mineral water in the study area.** The box represents the interquartile range (IQR), with the horizontal center line represents the median. Whiskers extend to the most extreme data points within 1.5 times the IQR, and individual points beyond the whiskers are outliers. The first and third quartile indicated by the top and bottom edge. Different lowercase letters (a, b, c.) on the boxplots denote significant differences between district medians as determined by Dunn's post-hoc test with Benjamini-Hochberg (BH) adjustment following a significant Kruskal-Wallis test (p < 0.05). **'a'** represents the group with the highest median concentration within each mineral.

p < 0.001), and Mo (r = −0.85, p < 0.01). Furthermore, in Gechi district, negative correlations were observed between EPG and serum Cu (r = −0.69, p < 0.05), Zn (r = −0.74, p < 0.05), and Mo (r = −0.69, p < 0.05). Moreover in Borecha district, negative correlations were observed between EPG and serum Cu (r = −0.75, p < 0.05) and Mo (r = −0.84, p < 0.01). Notably, Mn also showed a significant negative correlation in the overall analysis, while Mg presented a positive correlation only in Dabo district. Eventually, across the study districts, negative correlations were observed between EPG and serum Fe (r = −0.77, p < 0.01), Mn (r = −0.70, p < 0.01), Cu (r = −0.64, p < 0.05), and Zn (r = −0.80, p < 0.01).

### Negative binomial regression analysis of serum minerals on EPG

Univariate negative binomial regressions revealed the individual association of each serum mineral with EPG (Table 4). The dispersion parameter (Theta) for all models was estimated and found to be greater than 1 (ranging from

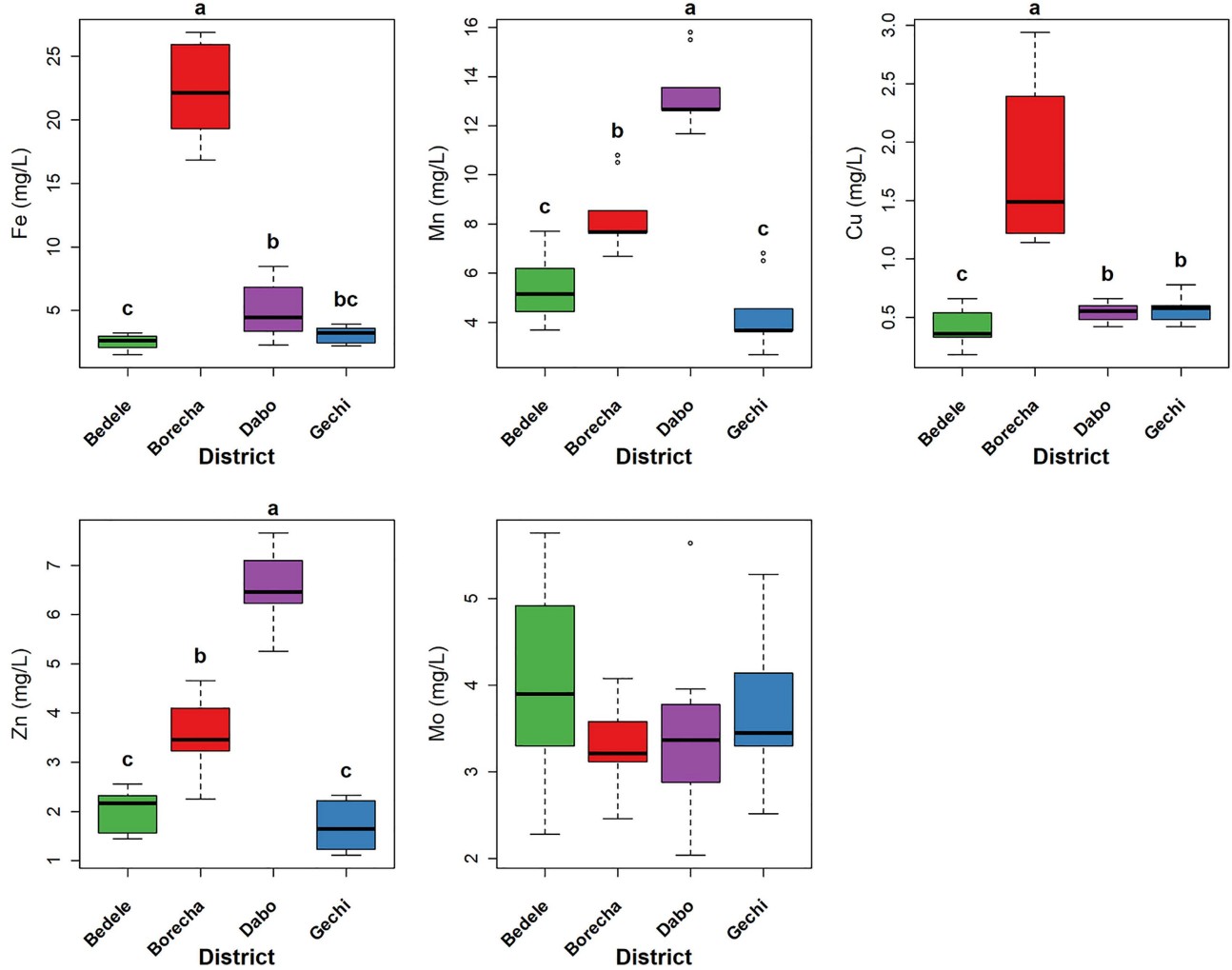

**Fig 4. Box plots of serum trace mineral concentrations in zebu cattle (*B. indicus*) grazing around and accessing *hora* mineral water in the study area.** The box represents the interquartile range (IQR), with the horizontal center line represents the median. Whiskers extend to the most extreme data points within 1.5 times the IQR, and individual points beyond the whiskers are outliers. The first and third quartile indicated by the top and bottom edge. Different lowercase letters (a, b, c.) on the boxplots denote significant differences between district medians as determined by Dunn's post-hoc test with Benjamini-Hochberg (BH) adjustment following a significant Kruskal-Wallis test (p < 0.05). **'a'** represents the group with the highest median concentration within each mineral.

11.04 to 23.08), confirming the presence of overdispersion in the EPG data and justifying the use of negative binomial regression over poisson regression. After applying the Benjamini-Hochberg (BH) adjustment to control for multiple comparisons, significant negative associations with EPG were observed for Ca, Mg, P, Fe, Mn, Cu, and Zn (p < 0.05). Na, K, S, and Mo did not show a statistically significant relationship with EPG after BH correction.

Multivariable stepwise negative binomial regression with the most parsimonious set of serum mineral predictors for EPG are shown in Table 5. The full model, including all mineral predictors, had an AIC of 375.94. The stepwise procedure selected a reduced model with a lower AIC of 368.61, indicating a superior balance between model fit and complexity. The stepwise multivariable negative binomial regression identified Fe, Cu, Zn, and Mo as the most robust predictors of EPG (Table 5). The final model, in log scale, was: log(EPG) = 5.92–0.011Fe - 0.139Cu - 0.100Zn - 0.089Mo. Variance Inflation

**Table 3. Correlation of serum mineral concentrations and mean fecal egg counts (EPG) of zebu cattle in the study districts.**

| Districts | | | Ca | Mg | Na | K | P | S | Fe | Mn | Cu | Zn | Mo |
|---|---|---|---|---|---|---|---|---|---|---|---|---|---|
| Bedele | EPG | Correlation coefficient | 0.40 | 0.16 | −0.30 | −0.51 | −0.53 | 0.01 | −0.76 | 0.27 | −0.92 | −0.65 | −0.07 |
| | | Significance level | ns | ns | ns | ns | ns | ns | * | ns | *** | * | ns |
| Dabo | EPG | Correlation coefficient | −0.22 | 0.84 | 0.07 | 0.08 | −0.28 | −0.18 | −0.98 | −0.30 | −0.89 | −0.94 | −0.85 |
| | | Significance level | ns | ** | ns | ns | ns | ns | *** | ns | *** | *** | ** |
| Gechi | EPG | Correlation coefficient | 0.51 | 0.25 | −0.46 | 0.42 | −0.60 | 0.29 | −0.57 | −0.53 | −0.69 | −0.74 | −0.69 |
| | | Significance level | ns | ns | ns | ns | ns | ns | ns | ns | * | * | * |
| Borecha | EPG | Correlation coefficient | −0.60 | −0.20 | 0.47 | 0.56 | −0.35 | 0.38 | −0.16 | −0.42 | −0.75 | −0.55 | −0.84 |
| | | Significance level | ns | ns | ns | ns | ns | ns | ns | ns | * | ns | ** |
| Overall | EPG | Correlation coefficient | −0.47 | −0.43 | 0.15 | 0.00 | −0.35 | 0.12 | −0.77 | −0.70 | −0.64 | −0.80 | −0.22 |
| | | Significance level | ns | ns | ns | ns | ns | ns | ** | ** | * | ** | ns |

*EPG = egg per gram; ns = non-significant (p > 0.05); * = p < 0.05; ** = p < 0.01; *** = p < 0.001.*

**Table 4. Univariate negative binomial regression results for serum minerals and EPG.**

| Minerals | Estimate | Std. Error | z value | Pr(>|z|) | Adjusted Pr(>|z|) (BH) | Significance | 95% CI |
|---|---|---|---|---|---|---|---|
| Ca | −0.0023 | 0.00057 | −4.040 | 5.4e-05 | 1.18e-04 | *** | −0.0034, −0.0012 |
| Mg | −0.0105 | 0.00356 | −2.935 | 0.0033 | 6.11e-03 | ** | −0.0169, −0.0036 |
| Na | 0.00006 | 0.00012 | 0.553 | 0.580 | 0.6381 | ns | −0.0001, 0.0003 |
| K | 0.00018 | 0.00141 | 0.126 | 0.900 | 0.8999 | ns | −0.0025, 0.0028 |
| P | −0.0047 | 0.00183 | −2.588 | 0.0096 | 0.0151 | ** | −0.0084, 0.0010 |
| S | 0.00019 | 0.00030 | 0.632 | 0.527 | 0.6381 | ns | −0.0004, 0.0007 |
| Fe | −0.0219 | 0.00486 | −4.499 | 6.8e-06 | 1.88e-05 | *** | −0.0310, −0.0125 |
| Mn | −0.0537 | 0.01034 | −5.196 | 2.0e-07 | 1.12e-06 | *** | −0.0734, −0.0338 |
| Cu | −0.2952 | 0.06183 | −4.773 | 1.8e-06 | 6.64e-06 | *** | −0.4113, −0.1750 |
| Zn | −0.1092 | 0.01775 | −6.153 | 7.6e-10 | 8.37e-09 | *** | −0.1429, −0.0751 |
| Mo | −0.0463 | 0.05646 | −0.820 | 0.412 | 0.5671 | ns | −0.1526, 0.0627 |
| Intercept | 5.2520 | 0.20913 | 25.11 | <2e-16 | – | *** | – |

*EPG = Eggs Per Gram; BH = Benjamini-Hochberg; Significance codes: *** p < 0.001, ** p < 0.01, * p < 0.05; ns = non-significant (p > 0.05); Values reported as < X.Ye-ZZ indicate p-values below the computational precision limit.*

**Table 5. Multivariable stepwise negative binomial regression results for EPG.**

| Predictor | Estimate | Std. Error | z value | Pr(>|z|) | Significance | VIF |
|---|---|---|---|---|---|---|
| Intercept | 5.9168 | 0.1105 | 53.55 | <2e-16 | *** | – |
| Fe | −0.0106 | 0.0052 | −2.04 | 0.0418 | * | 3.53 |
| Cu | −0.1391 | 0.0673 | −2.07 | 0.0387 | * | 3.45 |
| Zn | −0.1004 | 0.0116 | −8.69 | <2e-16 | *** | 1.03 |
| Mo | −0.0886 | 0.0265 | −3.34 | 0.0008 | *** | 1.04 |

*EPG = Eggs Per Gram; VIF = Variance Inflation Factor; Significance codes: *** p < 0.001, ** p < 0.01, * p < 0.05. Model AIC: 372.47; Dispersion parameter (Theta): 72.91.*

Factors (VIFs) for the predictors in the final model ranged from 1.03 to 3.53, indicating that multicollinearity was not a significant concern.

To visually assess the linearity assumption and identify influential observations for the final stepwise selected model (EPG ~ Fe + Cu + Zn + Mo), Component + Residual Plots (Cr Plots) were generated for each predictor (Fig 5). Component residual plots revealed a strong overall linear trend for each significant predictor, with minor non-linear deviations observed in some areas as shown in Fig 5. While linear terms were fitted for all variables, there are visual indications of non-linearity for Fe and Mo. However, the relationships for Cu and Zn appear more linear, although with minor deviations at the extremes, but these were not substantial enough to suggest a severe violation of the linearity assumption.

## Discussion

The observed overall GIP prevalence of 55.0% in zebu cattle accessing *hora* mineral water in this study presents a mixed picture compared to previous reports from free-ranging cattle in the region. While the GIP burden in current zebu cattle aligns with the majority of previous studies in similar free-ranging systems [41–43], it was notably lower than the higher prevalence reported by Tiele et al. [44] (67.2%) and Teshome et al. [45] (58.3%). This difference likely stems from variations in environmental conditions, management and feeding practices, with a pertinent contributing factor being the consistent access to mineral-rich sources like *hora* mineral water in our study area. This study identified a diverse range of GIP in zebu cattle, with *Strongyle*-type nematodes being the most prevalent group. This aligns with consistent documentation that nematodes are the major parasitic helminths in tropical and subtropical climates [46,47], known for causing

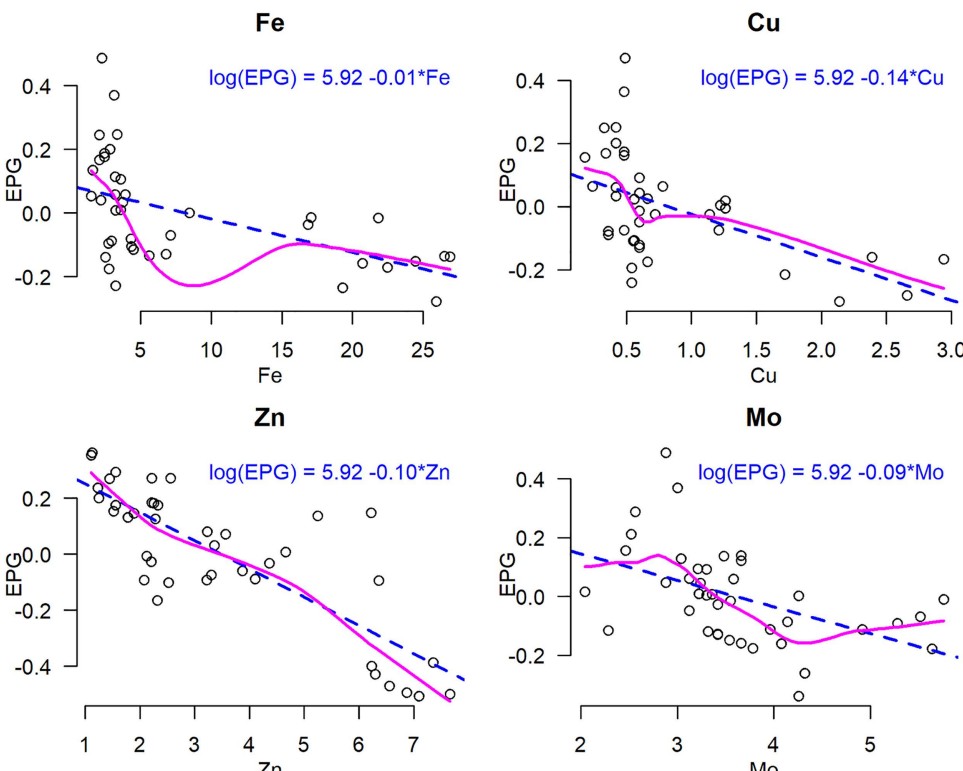

**Fig 5. Component-plus-residual plots for the final stepwise negative binomial regression model.** Each panel displays the relationship between the EPG and the indicated mineral (Fe, Cu, Zn, Mo), adjusted for other predictors in the model. The blue dashed line indicates a general negative linear trend, the magenta Loess curve suggests a distinct non-linear relationship. A blue equation string shows the log-linear relationship for each variable.

economic loss of free ranging zebu cattle by inducing anemia hindering growth [44,48]. *Paramphistomum spp.* and *Fasciola spp.* were also present as less frequent but significant trematode classes affecting zebu cattle in the study area. The overall mean EPG of 161.87 in zebu cattle routinely accessing *hora* mineral water were substantially lower compared to some reports from similar free-ranging cattle population in the region [12,49,50], and broadly consistent with light to moderate infection levels established by FAO [35] for grazing cattle in tropical region. The lower mean EPG values observed in this study suggesting that factors like the consistent access to *hora* mineral water in the study area might play a role in maintaining lighter infection levels.

Given our study's correlational design and the absence of a control group (cattle without access to *hora* mineral water), we acknowledge that our conclusions regarding any direct causal effect of *hora* water on parasite reduction remain correlational, and it is challenging to definitively ascertain if the observed low EPG is solely due to the water or other unmeasured environmental factors. Nevertheless, it is significant that both the overall mean EPG (161.87) for the study population and, for nematodes (101.59 EPG), consistently fall within the light to moderate infection range established by FAO [35] for grazing zebu cattle in tropical regions. Crucially, this study demonstrate significant internal variations within the *hora* accessing population that strongly support the proposed hypothesis and the potential influence of *hora* mineral water on parasite burden. The results showed a significantly higher overall mean EPG in Gechi district compared to Dabo and Borecha districts. This disparity indicates that cattle in Gechi district, despite having access to *hora* mineral water sources, may utilize them less frequently or consistently than cattle in Dabo and Borecha districts. This inferred differential access frequency is strongly corroborated by serum mineral analysis. Serum concentrations of macro-minerals (Ca, Mg, and P) and micro-minerals (Fe, Cu, and Zn) were significantly higher in cattle from Borecha district compared to other districts, indicating an enhanced mineral status. This improvement is highly likely due to their more consistent and frequent access to *hora* mineral water, whose analysis (Appendix 3 in S1 File) confirmed adequate levels of these essential minerals. This finding underscores the essential role of traditional practices mineral supplementation, such as providing access to *hora* during grazing, in maintaining livestock health where commercial mineral supplements may be limited. Traditional mineral licks such as salt licks, soil licks, and mineral-rich water have been documented in other pastoral systems, demonstrating their critical role in livestock health where commercial supplements are scarce [51–53].

The observed serum macro-mineral concentrations (Ca, Mg, Na, P, K, and S) in zebu cattle were generally within or exceeded marginal deficiency thresholds established by Suttle [54] (Appendix 2 in S1 File), indicating an adequate macro-mineral status. This contrasts with several studies reporting mineral deficiencies in grazing cattle under different management practices and geographical locations [55–58]. Such discrepancies are likely attributable to a combination of factors, including breed differences, variations in grazing management, and, importantly, the consistent access to mineral rich *hora* in the present study. The mineral-rich nature of *hora*, as supported by supplementary data, thus appears to significantly enhance the overall mineral status of zebu cattle, mitigating the risk of macro-mineral deficiency compared to cattle grazing in less mineral-rich areas. While serum micro-minerals (Fe, Mn, Zn) concentrations were found to be adequate, Cu was consistently below the established threshold for cattle [54] (Appendix 2 in S1 File) in all districts except Borecha, indicating potential deficiency. The sufficient concentrations of other micro-minerals, particularly Fe, may be contributing to this observed Cu deficiency, as high Fe intake can interfere with Cu absorption [59,60]. This observed Cu deficiency aligns with previous findings in zebu cattle (*Bos indicus*) [40] and yaks (*Bos grunniens*) [61], suggesting a common challenge in grazing cattle, even when other micro-minerals are present in adequate concentrations.

Statistical analysis further revealed a consistent inverse associations between serum mineral concentrations and GIP burden within the study population. Spearman correlation analyses demonstrated significant negative correlations between EPG and several serum micro-minerals concentrations (Fe, Mn, Cu, Zn, and Mo) across the study districts. Univariate negative binomial regressions, after controlling for multiple comparisons, further confirmed significant negative associations for Ca, Mg, P, Fe, Mn, Cu, and Zn with EPG. These consistent inverse relationship suggests that elevated serum concentrations of these minerals are associated with lower parasite burdens. Most critically, the multivariable

stepwise negative binomial regression analysis, which simultaneously considered all minerals, identified Fe, Cu, Zn, and Mo as the most robust predictors of EPG. The observed inverse relationships between serum Fe, Cu, Zn, and Mo concentrations and EPG strongly suggest their vital role in maintaining host health and resistance against GIP infection. Their robust selection in the final parsimonious model, alongside low Variance Inflation Factors (1.03–3.53) indicating minimal multicollinearity, indicates their collective importance in modulating GIP burden in cattle.

The consistent inverse relationships between serum essential minerals (Fe, Cu, Mo, and Zn) and EPG, evident in both correlation and regression analyses, reinforce the critical role of these minerals. These minerals are critical for various physiological processes, including immune function, which could directly impact an animal's ability to resist and clear parasitic infections [13,26,62]. For instance, previous research have been extensively reported he immunomodulatory effects of essential minerals, including Fe, Cu, Zn, Mo, and Mn [26,63,64]. Specifically, Zn plays a crucial role in T-cell proliferation, cytokine production, and antibody synthesis [15,65], and its deficiency is often linked to impaired immune responses against parasitic infections, which leads to increased susceptibility to disease [66]. Similarly, Cu and Fe are also vital for immune cell function and regulation of oxidative stress [67,68]. The observed strong negative correlations between serum concentrations of Fe, Cu, Zn, and Mn and EPG strongly support the hypothesis that adequate mineral status contributes to enhanced resistance against GIP infections in free ranging zebu cattle. Visual assessment of the final model's residuals (Component + Residual Plots, Fig 5) indicated overall linearity with minor deviations for Fe and Mo. The consistent negative correlations and the predictive power of the regression model, coupled with the observed district-level disparities linked to *hora* access frequency, present a powerful internal narrative. This creates a natural gradient of *hora* mineral exposure within the study population, acting as an implicit comparison that reinforces the premise that enhanced mineral availability, likely mediated by *hora* water consumption, is associated with a reduced parasite burden and potentially improved host resilience. The data obtained from this study is highly suggestive of *hora* mineral water playing a beneficial role by contributing to the mineral balance necessary for robust immune function against GIPs.

Although this study providing valuable insights, some limitations should be acknowledged. The natural environment of free-ranging animals introduces inherent variability. Confounding factors such as agro-ecological differences, grazing management practices, uncontrolled access to other mineral sources, animal-specific factors, forage mineral composition, and other potential health conditions could influence both serum mineral status and GIP burden, potentially confounding the observed correlations. The absence of comparison groups (cattle without access to *hora* mineral water) limits our ability to fully understand the specific effect of *hora* mineral water. It is also important to note that while this study is based on a cross-sectional analysis, which identifies strong associations, it does not establish a definitive causal relationship. The observed non-linearities for Fe and Mo are also a significant consideration, suggesting that future research should explore more flexible modeling techniques, such as generalized additive models (GAMs), various forms of spline regression, or polynomial contrast models, to more accurately capture these complex relationships.

## Conclusion

This study identified GIP as a significant challenge in zebu cattle accessing *hora* mineral water, with *Strongyle*-type nematodes being the predominant group. A key finding was the strong association between enhanced serum mineral concentrations, likely influenced by consistent access to mineral-rich *hora* water, and a reduced GIP burden. The result consistently suggest that adequate levels of essential minerals, particularly Fe, Cu, Zn, and Mo, contribute to improved host resistance against GIP infections, potentially by enhancing immune responses. Although this research provides valuable insights into the relationship between serum mineral status and parasite burden in naturally exposed cattle, its correlational design precludes the establishment of a direct causal relationship. Therefore, future research employing controlled feeding trials, ideally comparing *hora* mineral water access with non-access, alongside immunological assays is crucial to elucidate the precise mechanisms by which *hora* mineral water and its constituent minerals influence parasite resistance and to definitively establish a causal link between mineral status and parasite burden in grazing zebu cattle.

## Supporting information

**S1 File. Supplementary Appendix.** Lists of tables of descriptive statistics of analyzed data.
(DOCX)

**S2 File. Raw data of GIP prevalence and EPG of positive samples.**
(XLSX)

**S3 File. Raw data of mineral concentrations of *hora* mineral water.**
(XLSX)

**S4 File. Raw data of serum mineral concentration of zebu cattle used in this study.**
(XLSX)

## Acknowledgments

We thank to Buno Bedele Zone Agriculture Office and their field experts for their invaluable technical expertise and assistance in facilitating sample collection.

## Author contributions

**Conceptualization:** Ashenafi Miresa, Taye Tolemariam, Ellen S. Dierenfeld, Abebe Nigussie, Feyissa Begna.

**Data curation:** Ashenafi Miresa, Taye Tolemariam, Belay Duguma, Ellen S. Dierenfeld, Abebe Nigussie, Feyissa Begna.

**Formal analysis:** Ashenafi Miresa.

**Funding acquisition:** Ashenafi Miresa.

**Investigation:** Ashenafi Miresa.

**Methodology:** Ashenafi Miresa, Taye Tolemariam, Ellen S. Dierenfeld, Abebe Nigussie, Feyissa Begna.

**Project administration:** Ashenafi Miresa.

**Resources:** Ashenafi Miresa.

**Software:** Ashenafi Miresa.

**Supervision:** Ashenafi Miresa, Taye Tolemariam, Belay Duguma, Feyissa Begna.

**Validation:** Ashenafi Miresa, Taye Tolemariam, Belay Duguma.

**Visualization:** Ashenafi Miresa.

**Writing – original draft:** Ashenafi Miresa.

**Writing – review & editing:** Ashenafi Miresa, Taye Tolemariam, Belay Duguma, Ellen S. Dierenfeld, Abebe Nigussie, Feyissa Begna.

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
