## [Decision Letter · Decision Letter 0]

20 Feb 2025

PONE-D-25-01800Correlation of mineral status with gastrointestinal parasite burden in zebu (Bos indicus) cattle grazing around natural mineral water (hora) in southwestern Ethiopia.PLOS ONE

Dear Dr. Miresa,

Thank you for submitting your manuscript to PLOS ONE. After careful consideration, we feel that it has merit but does not fully meet PLOS ONE’s publication criteria as it currently stands. Therefore, we invite you to submit a revised version of the manuscript that addresses the points raised during the review process.

We look forward to receiving your revised manuscript.

Kind regards,

Bersissa Kumsa, DVM, MSc, PhD

Academic Editor

PLOS ONE

Journal Requirements:

“This research was supported by Jimma University College of Agriculture and Veterinary Medicine (grant number AgVmVm/M6/23) which provided funding for data collection and laboratory analysis.”

“We thank to Jimma University College of Agriculture and Veterinary Medicine for their financial support. We also extend our thanks to Buno Bedele Agriculture Office and field experts for their support during sample collection.”

“This research was supported by Jimma University College of Agriculture and Veterinary Medicine (grant number AgVmVm/M6/23) which provided funding for data collection and laboratory analysis.”

4. In the online submission form, you indicated that “The data presented in this study is available from the corresponding author upon reasonable request.“

6. We notice that your supplementary figures are uploaded with the file type 'Figure'. Please amend the file type to 'Supporting Information'. Please ensure that each Supporting Information file has a legend listed in the manuscript after the references list.

**Additional Editor Comments:**

The manuscript requires significant corrections on materials and method section as well as extensive edition on grammar, sentence structure, and language to ensure readability and adherence to the standards of journal.

Please address all the comments suggested by the two reviewers.

Reviewers' comments:

Reviewer's Responses to Questions

**Comments to the Author**

1. Is the manuscript technically sound, and do the data support the conclusions?

Reviewer #1: Partly

Reviewer #2: No

2. Has the statistical analysis been performed appropriately and rigorously? 

Reviewer #1: Yes

Reviewer #2: No

3. Have the authors made all data underlying the findings in their manuscript fully available?

Reviewer #1: Yes

Reviewer #2: Yes

4. Is the manuscript presented in an intelligible fashion and written in standard English?

Reviewer #1: No

Reviewer #2: No

5. Review Comments to the Author

Reviewer #1: The authors have examined only 45 animals from each district. The manuscript should provide a clear explanation of how this sample size was determined and the basis for selecting this specific number. Include statistical or logistical reasoning to justify the representativeness of the sample size.

The authors used 100x magnification to examine the eggs. Provide a justification for choosing this magnification instead of 10x or 40x, which are commonly used for egg examination. Clarify why 100x was deemed necessary and how it improved the accuracy or visibility of the eggs.

Add suitable references to support statements made in lines 92-102 and 124-134. Ensure that these references are relevant, up-to-date, and from credible scientific sources to strengthen the manuscript’s credibility.

Ensure that all scientific names throughout the text are italicized, including those mentioned in Figure 1. This is a standard convention in scientific writing.

The manuscript does not thoroughly discuss its limitations. Include a section addressing potential limitations such as:

Sample size constraints.

Reliance on specific districts and whether the findings can be generalized.

Other potential health and environmental factors that might influence parasite prevalence but were not considered in the study.

Some parts of the discussion lack clarity and depth. Revise this section to ensure that the results are interpreted comprehensively and placed in the context of existing literature. Provide more detailed insights into the implications of the findings.

The manuscript requires significant editing for grammar, sentence structure, and language to ensure readability and adherence to the standards of journal. Additionally, ensure consistency in formatting across all sections.

Reviewer #2: “Correlation of mineral status with gastrointestinal parasite burden in zebu (Bos indicus)

cattle grazing around natural mineral water (hora) in southwestern Ethiopia” aims to investigate the correlation between serum mineral status and gastrointestinal parasite burden in zebu cattle grazing around and accessing natural mineral water sources in southwestern Ethiopia. However, there are several issues with the manuscript that made me reject it for publication. The main ones are related to the material and methods. There are a series of non-controlled variables that can affect the results, which basically compare regions. For instance, lack of information and collection criteria about breed, age and physiological status from the animal side; pasture availability, forage composition, and “hora” water composition from the nutrition side. Definitely, methods are not appropriate to achieve the goals. Blood was left to coagulate. Hemolysis of serum samples is recognized to be the leading cause of preanalytical errors in clinical laboratories because it can significantly affect the mineral status of cattle serum, leading to inaccurate mineral profile assessments (doi: 10.3390/ani11123336). Also, being qualitative, parasitological analysis is not suitable to establish any correlation with nutrition because it indicates the prevalence and not the parasite burden. The manuscript looks like more an attempt to assess mineral status and gastrointestinal parasite in zebu cattle in southwestern Ethiopia than a research paper to me. With the right methodologies, it could be a research note. Finally, it is inaccurate to analyze correlations with numerous uncontrollable variables.

6. PLOS authors have the option to publish the peer review history of their article (what does this mean? ). If published, this will include your full peer review and any attached files.

**Do you want your identity to be public for this peer review?** For information about this choice, including consent withdrawal, please see our Privacy Policy .

Reviewer #1: **Yes: ** Hafiz Muhammad Rizwan

Reviewer #2: No

---

## [Author Response · Author response to Decision Letter 1]

14 Mar 2025

I have no comments for specific reviewer and editors. I have already submitted and uploaded documents that responds to each point raised by the academic editor and reviewer(s) along with revision.

---

## [Decision Letter · Decision Letter 1]

19 Jun 2025

PONE-D-25-01800R1Correlation of serum mineral concentrations with gastrointestinal parasite burden in zebu cattle grazing around "hora" mineral water in southwestern EthiopiaPLOS ONE

Dear Dr. Miresa,

Thank you for submitting your manuscript to PLOS ONE. After careful consideration, we feel that it has merit but does not fully meet PLOS ONE’s publication criteria as it currently stands. Therefore, we invite you to submit a revised version of the manuscript that addresses the points raised during the review process.

We look forward to receiving your revised manuscript.

Kind regards,

Bersissa Kumsa, DVM, MSc, PhD

Academic Editor

PLOS ONE

Journal Requirements:

Additional Editor Comments:

Dear Authors,

The reviewers have completed their evaluation of your manuscript. I encourage you to revise and resubmit your work, ensuring that all reviewer comments are thoroughly addressed. Please incorporate the feedback carefully and provide a detailed, point-by-point response that clearly outlines every change made in response to the reviewers’ suggestions.

In addition, kindly correct all typographical and grammatical errors, and ensure that the manuscript is prepared in full compliance with the journal’s formatting and submission guidelines.

We look forward to receiving your revised submission

Reviewers' comments:

Reviewer's Responses to Questions

**Comments to the Author**

1. If the authors have adequately addressed your comments raised in a previous round of review and you feel that this manuscript is now acceptable for publication, you may indicate that here to bypass the “Comments to the Author” section, enter your conflict of interest statement in the “Confidential to Editor” section, and submit your "Accept" recommendation.

Reviewer #3: (No Response)

2. Is the manuscript technically sound, and do the data support the conclusions?

Reviewer #3: Partly

3. Has the statistical analysis been performed appropriately and rigorously? 

Reviewer #3: Yes

4. Have the authors made all data underlying the findings in their manuscript fully available?

Reviewer #3: Yes

5. Is the manuscript presented in an intelligible fashion and written in standard English?

Reviewer #3: Yes

6. Review Comments to the Author

Reviewer #3: (No Response)

7. PLOS authors have the option to publish the peer review history of their article (what does this mean? ). If published, this will include your full peer review and any attached files.

**Do you want your identity to be public for this peer review?** For information about this choice, including consent withdrawal, please see our Privacy Policy .

Reviewer #3: No

---

## [Author Response · Author response to Decision Letter 2]

4 Jul 2025

Dear Dr. Kumsa and Editorial Team,

Thank you for the opportunity to revise our manuscript, "Correlation of serum mineral concentrations with gastrointestinal parasite burden in zebu cattle grazing around 'hora' mineral water in southwestern Ethiopia" (PONE-D-25-01800R1).

We have carefully addressed all points raised by the Academic Editor and Reviewer #3. This revision includes:

We have thoroughly reviewed the manuscript for technical robustness and clarity.

We clarified our exploratory correlational approach and rationale for studying a single population with Hora access.

We corrected the reporting of strongyle-type nematode eggs to acknowledge limitations of morphological identification without culture, ensuring accuracy (e.g., now reported as 'strongyle-type nematode eggs').

We provided explicit details on the application of Pearson Chi-squared, Kruskal-Wallis, and Spearman's rank correlation tests.

We added details on practical measures taken during sample collection.

We included a study site map.

We ensured the reference list is complete, correct, and free of retracted articles.

We corrected all typographical and grammatical errors and ensured full compliance with journal guidelines.

A detailed point-by-point response is provided in the 'Response to Reviewers' file. We believe these revisions have significantly improved the manuscript's clarity, accuracy, and overall quality.

Sincerely,

Ashenafi Miresa Kenea (Corresponding Author)

---

## [Decision Letter · Decision Letter 2]

16 Jul 2025

PONE-D-25-01800R2Association between serum mineral concentrations and gastrointestinal parasite burden in zebu cattle accessing 'hora' mineral water in southwestern EthiopiaPLOS ONE

Dear Dr. Miresa,

Thank you for submitting your manuscript to PLOS ONE. After careful consideration, we feel that it has merit but does not fully meet PLOS ONE’s publication criteria as it currently stands. Therefore, we invite you to submit a revised version of the manuscript that addresses the points raised during the review process.

We look forward to receiving your revised manuscript.

Kind regards,

Bersissa Kumsa, DVM, MSc, PhD

Academic Editor

PLOS ONE

Journal Requirements:

Additional Editor Comments (if provided):

Dear Authors,

The reviewers have completed their evaluation of your manuscript. I encourage you to revise and resubmit your work, ensuring that all reviewer comments are thoroughly addressed. Please incorporate the feedback carefully and provide a detailed, point-by-point response that clearly outlines every change made in response to the reviewers’ suggestions.

In addition, kindly correct all typographical and grammatical errors, and ensure that the manuscript is prepared in full compliance with the journal’s formatting and submission guidelines.

We look forward to receiving your revised submission.

Reviewers' comments:

Reviewer's Responses to Questions

**Comments to the Author**

1. If the authors have adequately addressed your comments raised in a previous round of review and you feel that this manuscript is now acceptable for publication, you may indicate that here to bypass the “Comments to the Author” section, enter your conflict of interest statement in the “Confidential to Editor” section, and submit your "Accept" recommendation.

Reviewer #3: All comments have been addressed

2. Is the manuscript technically sound, and do the data support the conclusions?

Reviewer #3: Yes

3. Has the statistical analysis been performed appropriately and rigorously? 

Reviewer #3: Yes

4. Have the authors made all data underlying the findings in their manuscript fully available?

Reviewer #3: Yes

5. Is the manuscript presented in an intelligible fashion and written in standard English?

Reviewer #3: Yes

6. Review Comments to the Author

Reviewer #3: Data analysis:

Now the data analysis portion is well explained. One comment is that the authors said “Subsequently, a full multivariate model was constructed including all minerals as predictors, and stepwise regression using the Akaike Information Criterion (AIC) was then performed to select the most parsimonious model”. It is a multivariable model not multivariate model. Multivariate is when more than one outcome variable is modeled simultaneously but in this case it is more than one explanatory variables which are used to model one outcome variable. This needs to be corrected.

Results:

The districts are dominantly found in mid to lowland agro ecology and similar tropical weather conditions (from the study site description). Hence, how did the authors identify nematodes such as Nematodirus and Ostertagia? Their identification lacks reliability.

7. PLOS authors have the option to publish the peer review history of their article (what does this mean? ). If published, this will include your full peer review and any attached files.

**Do you want your identity to be public for this peer review?** For information about this choice, including consent withdrawal, please see our Privacy Policy .

Reviewer #3: **Yes: ** Teshale Sori

---

## [Author Response · Author response to Decision Letter 3]

19 Jul 2025

Authors’ response to editor and reviewers’ comments

Thank you for the opportunity to revise and resubmit our manuscript, “Association between serum mineral concentrations and gastrointestinal parasite burden in zebu cattle accessing 'hora' mineral water in southwestern Ethiopia" (PONE-D-25-01800R2). We appreciate the valuable feedback from the reviewers, which has significantly improved the quality of our manuscript. We have carefully addressed all comments raised by the academic editor and the reviewers, incorporating the suggested changes into the revised manuscript. A marked-up copy highlighting all revisions is provided as 'Revised Manuscript with Track Changes', and a clean version is provided as 'Manuscript(3)'.

In addition to addressing the specific comments below, we have also made a significant change to the presentation of our results. To enhance the clarity and immediate understanding of our key findings for the readers, we have modified and moved Table 4 and Table 5 from supporting information into the main body text of the manuscript. We believe that presenting this information directly within the main text provides a more comprehensive and accessible narrative of our research, allowing readers to readily grasp the nuances of our findings without needing to refer to supplementary files.

Our detailed point-by-point responses to all comments are presented below.

We really appreciate your help.

Ashenafi Miresa Kenea (corresponding author)

July 19, 2025

Academic Editor Comments:

The reviewers have completed their evaluation of your manuscript. I encourage you to revise and resubmit your work, ensuring that all reviewer comments are thoroughly addressed. Please incorporate the feedback carefully and provide a detailed, point-by-point response that clearly outlines every change made in response to the reviewers’ suggestions. In addition, kindly correct all typographical and grammatical errors, and ensure that the manuscript is prepared in full compliance with the journal’s formatting and submission guidelines. We look forward to receiving your revised submission.

Author Response: We thank the Academic Editor for this guidance. We have thoroughly reviewed our manuscript to ensure all previous reviewer comments are thoroughly addressed, as detailed in the previous revision and now in response to the current feedback. We have made a concerted effort to identify and correct all typographical and grammatical errors throughout the manuscript. We have also carefully checked and ensured full compliance with PLOS ONE’s formatting and submission guidelines, including the reference list.

Reviewers' comments:

Reviewer's Responses to Questions

Comments to the Author

1. If the authors have adequately addressed your comments raised in a previous round of review and you feel that this manuscript is now acceptable for publication, you may indicate that here to bypass the “Comments to the Author” section, enter your conflict of interest statement in the “Confidential to Editor” section, and submit your "Accept" recommendation.

Reviewer #3: All comments have been addressed

Author Response: We sincerely wish to thank Reviewer #3 for their time and his positive feedback. We are pleased to hear that all your comments have been addressed to your satisfaction.

2. Is the manuscript technically sound, and do the data support the conclusions?

Reviewer #3: Yes

Author Response: Thank you for your valuable feedback and for confirming the technical soundness of our manuscript and that our data support the conclusions. We appreciate your positive assessment.

3. Has the statistical analysis been performed appropriately and rigorously?

Reviewer #3: Yes

Author Response: Thank you for your valuable feedback and for confirming our statistical analysis was performed appropriately and rigorously. We appreciate your positive feedback.

4. Have the authors made all data underlying the findings in their manuscript fully available?

Reviewer #3: Yes

Author Response: Thank you for your review and for confirming that we've made all underlying data fully available as per the PLOS Data Policy. We appreciate your positive assessment.

5. Is the manuscript presented in an intelligible fashion and written in Standard English?

Reviewer #3: Yes

Author Response: Thank you for your time to review and for confirming that our manuscript was presented in an intelligible fashion and written in standard English. We appreciate your positive assessment on this important aspect.

6. Review Comments to the Author

Reviewer #3: Data analysis:

Now the data analysis portion is well explained. One comment is that the authors said “Subsequently, a full multivariate model was constructed including all minerals as predictors, and stepwise regression using the Akaike Information Criterion (AIC) was then performed to select the most parsimonious model”. It is a multivariable model not multivariate model. Multivariate is when more than one outcome variable is modeled simultaneously but in this case it is more than one explanatory variables which are used to model one outcome variable. This needs to be corrected.

Authors' Response: We thank the reviewer for this important clarification regarding statistical terminology. We concur that since our analysis involved modeling a single outcome variable (EPG) with multiple explanatory (mineral) variables, the appropriate term is "multivariable model." We have corrected all instances of "multivariate model" to "multivariable model" in the methodology and results sections of the manuscript to ensure accuracy.

Results:

The districts are dominantly found in mid to lowland agro ecology and similar tropical weather conditions (from the study site description). Hence, how did the authors identify nematodes such as Nematodirus and Ostertagia? Their identification lacks reliability.

Authors' Response: Thank you for your valuable feedback regarding the identification of nematode eggs (Nematodirus and Ostertagia) given the agro-ecological setting. We agree that relying solely on egg morphology for definitive species identification in the Strongyle-type group can be limiting, especially given the morphological similarities among many nematode eggs.

While our initial identification was based on the extensive experience of our laboratory experts, we acknowledge that a more scientifically rigorous approach, such as larval culture or molecular techniques, is essential for absolute confirmation at the genus or species level. We recognize that even highly experienced individuals can face challenges in differentiating between morphologically similar strongyle eggs.

Upon reviewing your comments and aligning with current scientific consensus, we've revised our manuscript. We now report these morphologically similar eggs as "Strongyle-type eggs" throughout the text.

We have made the necessary changes throughout the manuscript to reflect this revised reporting. We apologize for any ambiguity or inaccuracy in our previous reporting and appreciate you guiding us toward this important correction.

---

## [Editor Report · Decision Letter 3]

22 Jul 2025

Association between serum mineral concentrations and gastrointestinal parasite burden in zebu cattle accessing 'hora' mineral water in southwestern Ethiopia

PONE-D-25-01800R3

Dear Dr. Ashenafi,

We’re pleased to inform you that your manuscript has been judged scientifically suitable for publication and will be formally accepted for publication once it meets all outstanding technical requirements.

Kind regards,

Bersissa Kumsa, DVM, MSc, PhD

Academic Editor

PLOS ONE

Additional Editor Comments (optional):

We are pleased to inform you that your manuscript has been accepted for publication in the journal.

Please ensure that all typographical and grammatical errors are corrected prior to publication, and that the manuscript fully complies with the journal’s formatting and submission guidelines.

We look forward to receiving your future submissions and continuing our collaboration
---

## [Editor Report · Acceptance letter]

PONE-D-25-01800R3

PLOS ONE

Dear Dr. Miresa,

I'm pleased to inform you that your manuscript has been deemed suitable for publication in PLOS ONE. Congratulations! Your manuscript is now being handed over to our production team.

Kind regards,

on behalf of

Professor Bersissa Kumsa

Academic Editor

PLOS ONE